# Syngas from Reforming Methane and Carbon Dioxide on Ni@M(SiO_2_ and CeO_2_)

**DOI:** 10.3390/nano14231877

**Published:** 2024-11-22

**Authors:** Derun Hua, Jian Li, Xiaowen Guo, Xinning Lu, Hao Ding, Rengui He

**Affiliations:** Key Laboratory of Functional Materials Chemistry, Gannan Normal University, Ganzhou 341003, Chinaxinninglu@gnnu.edu.cn (X.L.); dhao2021@126.com (H.D.);

**Keywords:** reforming, Core@Shell material, MOF

## Abstract

The accumulation of greenhouse gasses (CH_4_ and CO_2_) results in an increase in the temperature of the atmosphere. The conversion of greenhouse gasses into chemicals and fuels with high added value benefits not only the environment but also energy development. A promising and well-studied process is the reforming of methane, where CH_4_ and CO_2_ are converted into syngas (CO and H_2_). However, catalysts hinder the development of the process. In this paper, we investigate the conversion of CH_4_ and CO_2_ into syngas using a thermal conversion method. The catalysis performance was evaluated by reforming methane. Ni-based catalysts were prepared by different methods. All prepared catalysts were characterized (XRD, HRTEM et al.), and the process of reforming carbon dioxide–methane was carried out in a fixed bed reactor under atmospheric pressure and a high temperature. Ni(M) @CeO_2_ is one of the most popular options due to the role of CeO_2_. The deposition of coke in Ni-based catalysts was investigated.

## 1. Introduction

The continuous consumption of non-renewable sources results in an increase in CO_2_ in the atmosphere, which results in the destruction of balance in the natural climate. Therefore, mitigating CO_2_ pollution has become an outstanding area of research. In view of this, researchers have investigated several strategies based on the development of catalysts. The catalytic conversion of CO_2_ into value-added fuels and chemicals has become one of the most practical CO_2_ mitigation approaches. So far, research on catalytic CO_2_ conversion has focused on electrochemical [1] and thermochemical approaches [2]. Apart from carbon dioxide, the high emission of CH_4_ is responsible for the temperature increase; therefore, it is urgent to control these emissions and convert CO_2_ and CH_4_ into other useful substances. Against this background, the dry reforming of CH_4_ and CO_2_ has attracted much attention because syngas is a prospective feedstock to produce chemical materials (see Figure 1).

The dry reforming of methane (DRM, $CO_{2}+CH_{4}=2CO+2H_{2}$) is one of the most promising and well-researched processes to mitigate global warming as it simultaneously converts the two most abundant greenhouse gasses (CH_4_ and CO_2_) into syngas (CO and H_2_), which is used not only as a gaseous fuel but also as a feedstock for high-value production, such as methyl alcohol and ethylene [3,4].

In this process, the deposition of carbon on the surface and the sintering of active metal particles are two major problems that lead to deactivation [5]. Despite precious metal-based catalysts are highly active, they undergo less coking and are high stable, they are not available and have high cost [6,7]. In comparison, Ni-based catalysts are more promising candidates for the DRM reaction as they are easily accessible.

Ni-based catalysts, with their fast conversion rates, availability, and low cost, are industrially accepted for dry reforming reactions [8] and frequently studied. Apanee et al. investigated the catalytic performance of Ni based on zeolites; the authors reported that Ni/zeolite Y showed better catalytic performance [9]. Kaydouh et al. reported that a high dispersion of nickel and cerium inside SBA-15 was achieved using the “two-solvent” deposition method [10]. It follows that the preparation method and carriers have a direct influence on the dispersion of nickel species. Despite their low cost and catalytic activity, Ni-based catalysts are more susceptible to deactivation by carbon deposition than noble metal-based catalysts [5,11]. Therefore, researchers have recently tried to improve the catalytic properties of Ni-based catalysts. Bimetallic catalysts show better performance than Ni-based catalysts [12,13,14]. The catalytic performance of catalysts has also been improved by doping promoters [15,16], which create new active sites and prevent carbon formation [17,18].

In addition, the performance of the catalysts was improved by inhibiting carbon deposition [5,17,19]. It was reported that cerium oxide can prevent the carbon deposition of catalysts. The reducibility and the total basicity of the catalysts were improved by the addition of Ce, which was responsible for the improvement in the catalytic activity [2,18,20,21,22].

Recently, much attention has been paid to materials with core/shell structures in which the active metals are confined to narrow regions and can effectively hinder the aggregation of active nanoparticles by the confinement effect of the shell [23,24]. Ji et al. reported that a novel Ni@SiO_2_ yolk–shell nanoreactor is used as a model catalyst for the steam methane reforming reaction and is superior to Ni/MCF catalysts and comparable to modern commercial catalysts [23]. Kang et al. synthesized Ni supported on Al_2_O_3_ and MgO–Al_2_O_3_ with core/shell structures with stability [25]. Li et al. reported that Ni nanoparticles (NPs) with a narrow size distribution was encapsulated by meso- and microporous silica [26]. 

The great attention paid to core/shell structures is due to the controllability of the components of the shell. It is difficult to synthesize such composites, and intensive research is still ongoing [27,28].

In this work, we aim to investigate the effects of reaction conditions on the performance of catalysts and coke. We are interested in elucidating the role of Ce in the dry reforming of methane, especially its influence on carbonaceous deposition.

## 2. Results and Discussion

### 2.1. Characterization of Catalysts

The TEM images show that Ni nanoparticles with different sizes were randomly distributed (Figure 2). In particular, the size of the relatively larger Ni particles was about 20 nm, and that of the relatively smaller Ni particles was about 7 nm, and the Ni particle size distribution centered at 10 nm. A Ni (110) crystal plane corresponding to a lattice spacing of about 0.182 nm was found.

### 2.2. Test of Catalysts

The dry reforming of methane and carbon dioxide is not spontaneous below 400 °C and at atmospheric pressure as it is a highly endothermic reaction; therefore, the reaction temperature is set above 700 °C.

An effect of the pretreatment temperature on the catalysts was shown in Figure 3. The observed catalytic conversions generally increased with the pretreatment temperature, and the maximum conversion was observed at 700 °C. Then, the conversions decreased with the increase in the pretreatment temperature, indicating that Ni^x+^ can be completely reduced to Ni^0^ below 700 °C; moreover, the Ni particles were aggregated above 700 °C. In general, the H_2_/CO ratio seems to be related to the activity of the catalysts. As confirmed in the current study, the H_2_/CO ratio depends on the degree of Ni reduction. The H_2_/CO ratio was found to be less than 1, which is due to the occurrence of the reverse water–gas shift reaction ($CO_{2}+H_{2}=CO+H_{2}O$).

The stability of four catalysts was determined at 750 °C. Figure 4 shows the conversion of methane, carbon dioxide, and the molar ratio H_2_/CO with time on stream (9 h) over four catalysts. As can be seen in the figure, there was no change in conversion for three catalysts (Ni(M)@SiO_2_, Ni@CeO_2_, and Ni(M)@CeO_2_), indicating that the three catalysts hold good duration, but there was an outstanding change in the H_2_/CO ratio for Ni@SiO_2_. The Ni@SiO_2_ catalyst showed a much worse duration than the others, while Ni(M)@CeO_2_ exhibited the best stability among the four catalysts. In addition, we found that the conversion of CO_2_ was higher than that of methane for the equation $CO_{2}+H_{2}=CO+H_{2}O$. After a duration of 9 h, the CH_4_ and CO_2_ conversions showed analogous trends and remained stable, while the H_2_/CO molar ratio of the Ni@SiO_2_ catalyst decreased markedly from 0.86 to 0.8. According to the above results, the performance of the catalysts was affected by the pretreatment temperature and the composition of the shell. According to [29], the decomposition of methane and the Boudouard reaction are the main causes for the formation of carbon during the dry reforming reaction. At low temperatures and a high pressure, the Boudouard reaction dominates for the exothermic reaction and is a lower amount of gas moles. In contrast, at a high temperature and a low pressure, methane cracking dominates for the endothermic reaction and the formation of more gas moles. In this study, carbon deposition from methane cracking was predominant.

Figure 5 shows the XRD data of the four samples prepared with various methods. As can be seen in Figure 5, the peak at 43.3° (visible in Figure 5) was identified for all samples and assigned to (202) NiO nanocrystals. The peak at 37.2° was only identified for Ni@SiO_2_ and Ni@CeO_2_ and assigned to (021) NiO nanocrystals. Ni° was oxidized to Ni^2+^ after the reaction. In addition, the peak at 37.2° was not identified for Ni(M) @SiO_2_ and Ni(M)@CeO_2_ prepared with MOFs (Appendix A), which can be attributed to the low nickel content or high dispersion of NiO particles. The peaks at 44.5°, 51.8°, and 76.3° refer to the Ni^0^ species. The XRD patterns of as-synthesized Ni(M)@SiO_2_ and Ni(M)@CeO_2_ show the existence of a distinct fluorite-like oxide structure of CeO_2_ (JCPDS 34-0394, space group Fm3m). There are eight diffraction peaks centered at 28.5°, 33.1°, 47.5°, 56.3°, 59.1°, 69.4°, 76.7°, and 79.1°. These peaks correspond to (111), (200), (220), (311), (222), (400), (331), (420), and (422) of the fcc structure, respectively [30].

The carbon deposition of the catalysts was determined by a thermal gravity analysis (TGA). As can be seen in Figure 6, there is no mass-loss step associated with the evaporation of water for the four catalysts. Weight loss was found above 150 °C and decreased as follows: Ni@SiO_2_ > Ni(M)@SiO_2_ > Ni@CeO_2_ > Ni(M)@CeO_2_. The quantitative coke contents of the spent Ni@SiO_2_, Ni(M)@SiO_2_, Ni@CeO_2_, and Ni(M)@CeO_2_ were determined to be 12.41, 8.26, 5.11, and 1.72 wt.%, respectively. Therefore, carbon deposition over Ni@CeO_2_ and Ni(M)@CeO_2_ is effectively inhibited for the redox nature of the CeO_2_ shell.

The surface structure of the reforming catalyst changed after carbonization. Therefore, XPS characterization was performed. As shown in Figure 7, the Ni 2p (Figure 7A) and Ce 3d (Figure 7B) XPS spectra of the spent Ni@CeO_2_ and Ni(M)@CeO_2_ are shown. The peak at 856.6 eV and the satellite peak at 861 eV were attributed to Ni^2+^ because Ni^0^ was oxidized to Ni^2+^ after the reaction. The peak at 852.4 eV and 859.6 eV was attributed to Ni^0^. According to the literature [31,32], the spectra were deconvoluted into many peaks, Ce 3d_5/2_ was tagged as v contributions, and Ce 3d_3/2_ was tagged as u contributions. The u″, u, and v‴ peaks were attributed to Ce^4+^, while the u′ and u^0^ peaks were attributed to Ce^3+^. The coexistence of Ce^4+^ and Ce^3+^ reveals the special property (redox) of the CeO_2_ shell, which facilitates the oxidation of the deposited carbon with the participation of lattice oxygen. CeO_2_ is a suitable material for reforming methane due to its high specific surface area and abundant oxygen vacancies.

## 3. Experimental Procedure

### 3.1. Catalyst Synthesis 

All used chemicals were not further purified.

(1)Synthesis of Ni nanoparticles [33]

Ni(acac)_2_ (1.00 g; 3.90 mmol) was added to 39.0 mmol of oleylamine (10.4 g; 5 equiv) and 3.12 mmol of trioctylphosphine (1.15 g, 0.4 equiv). The mixture was degassed at 100 °C and then heated at 220 °C for 2 h under a nitrogen atmosphere. The mixture was cooled to room temperature and centrifuged after adding 40 mL of acetone to obtain a black product. The nanoparticles were redispersed in a mixture of hexane and ethanol (molar ratio of 3:1).

(2)Synthesis of Ni@CeO_2_(SiO_2_) [34]

Ni nanoparticles were coated with silicon dioxide and cerium dioxide. The suspension of Ni nanoparticles (15 mL) was mixed with 20 mL P123 and 3 mL ammonia (30 wt%) to form a microemulsion and stirred for 60 min. An amount of 1.5 mL of TEOS or cerium nitrate (2 mL; 1 mol/L) was added using a syringe pump with a flow rate of 1 mL·h^−1^. After 96 h of reaction, the Ni@SiO_2_ nanoparticles (Ni@CeO_2_ nanoparticles) were separated and washed using ethanol, dried in air, and calcined at 650 °C for 3 h. 

(3)Synthesis of Ni-BTC [35]

1,3,5-Benzenetricarboxylic acid (0.105 g) and sodium hydroxide (0.06 g) were dissolved in 30 mL of deionized water; the resulting mixture was stirred at room temperature for 30 min. Then, Ni (NO_3_)_2_∙6H_2_O (0.29 g) was added to the above solution and stirred well. Finally, the reaction mixture was added to a 50 mL stainless steel reactor, sealed, and heated in an oven to 130 °C for 72 h. After the reaction, it was naturally cooled to room temperature, and the blue Ni-BTC crystals were collected from the final reaction mixture by filtration and air-dried at room temperature. 

(4)Synthesis of Ni(M)@CeO_2_ and Ni(M)@SiO_2_ based on metal–organic frameworks (MOFs) [36,37]

MOF powder (100 mg) was dispersed in ammonium hydroxide (5 wt%) aqueous solution. Then, a certain amount of octadecyl trimethoxy silane or cerium nitrate was added to the above mixture with a micropump (2 mL/h). The above mixture was kept at 80 °C for 72 h, MOF@CeO_2_ and MOF@SiO_2_ were synthesized and calcined at 600 °C, and Ni(M) @CeO_2_ and Ni(M)SiO_2_ were obtained. 

### 3.2. Characterization of Catalyst

The X-ray powder diffractograms of the catalysts were recorded with a Bruker D8 ADVACE diffractometer using CuKa (1.5406 Å) radiation in the range of 5–85° with a scanning rate of 1°/min. High-resolution transition electron microscopy (HR-TEM) images were obtained with a Tecnai G2F20 Super-twin (FEI) microscope (FEI Corporation, Hillsboro, OR, USA). Thermogravimetric analysis (TGA) was carried out on a TGAQ50 (TA INSTRUMENTS). To perform the TGA test, 4 mg of spent catalyst was loaded, and the temperature was increased from 25 to 800 °C with a temperature ramp of 10 °C·min^−1^ in an air flow.

### 3.3. Evaluation of Catalytic Performance 

The evaluation of catalytic performance was carried out in a fixed bed quartz reactor with an inner diameter of 5 mm. Typically, 100 mg of catalysts was loaded into the reactor and reduced with mixed gas (H_2_/N_2_ = 1:5; 10 mL/min) for 30 min at 650 °C. In all studies, the feed gas was used with a constant of V(CH_4_):V (CO_2_):V (Ar) = 1:1:2, where Ar was used as the carrier gas. The catalytic reaction was evaluated under GHSV = 6000 mL·g^−1^⋅h^−1^. The composition of exhaust gas was determined using an online GC system (Tian Mei GC-7980, Quanzhou, China) equipped with a TDX-01 column and a TCD. The conversion ($C_{{CH}_{4}}\;and\;C_{{CO}_{2}}$) and selectivity ($S_{CO}\;and\;S_{H_{2}}$) were calculated using the following equations:(1)$$C_{{CH}_{4}}=\frac{F_{CH_{4},in}-F_{CH_{4},out}}{F_{CH_{4},in}}\times 100\%$$
(2)$$C_{{CO}_{2}}=\frac{F_{{CO}_{2},in}-F_{{CO}_{2},out}}{F_{{CO}_{2},in}}\times 100\%$$
(3)$$S_{CO}=\frac{F_{CO,out}}{F_{CH_{4},in}+F_{{CO}_{2},in}}\times 100\%$$
(4)$$S_{H_{2}}=\frac{F_{H_{2},out}}{F_{CH_{4},in}}\times 100\%$$
(5)$$\frac{H_{2}}{CO}=\frac{F_{H_{2},out}}{F_{CO,out}}$$

## 4. Conclusions

The aim of this project was to investigate the suitability of Ni@M catalysts with different compositions and preparation methods as catalysts for dry reforming. It was important to show the effects of the shell on the activity and stability of the catalysts.

Therefore, various catalysts, such as Ni@SiO_2_, Ni(M)@SiO_2_, Ni@CeO_2_, and Ni(M)@CeO_2_, were synthesized and characterized using numerous technologies (HRTEM, XRD, etc.). The characterized results reveal the structures of the catalysts.

In summary, the Ni(M)@/CeO_2_ catalyst exhibited better stability than the other catalysts. The CO_2_ conversion and CH_4_ conversion were 85% and 87%, respectively. The TGA profile confirms that less carbon was deposited on the Ni(M)@CeO_2_ catalyst. The XPS characterizations show that the Ni(M)@CeO_2_ catalyst possessed a stronger redox property and enriched surface basicity, which may improve the coke resistance of the Ni(M)@CeO_2_ catalyst.

## Figures and Tables

**Figure 1 nanomaterials-14-01877-f001:**
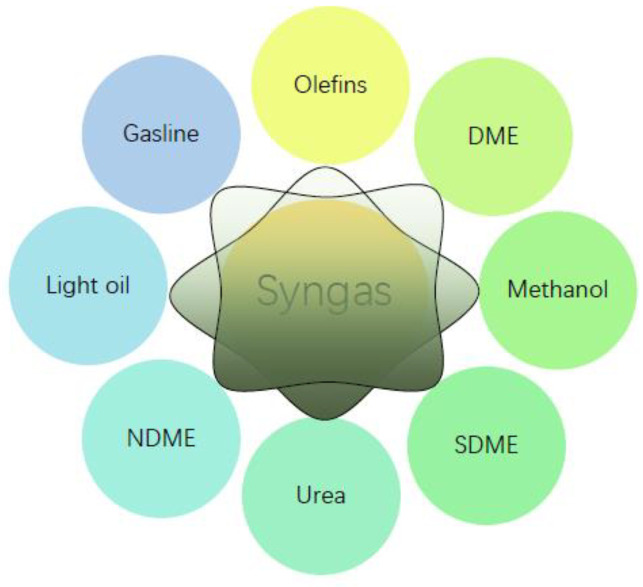
The utilization of syngas.

**Figure 2 nanomaterials-14-01877-f002:**
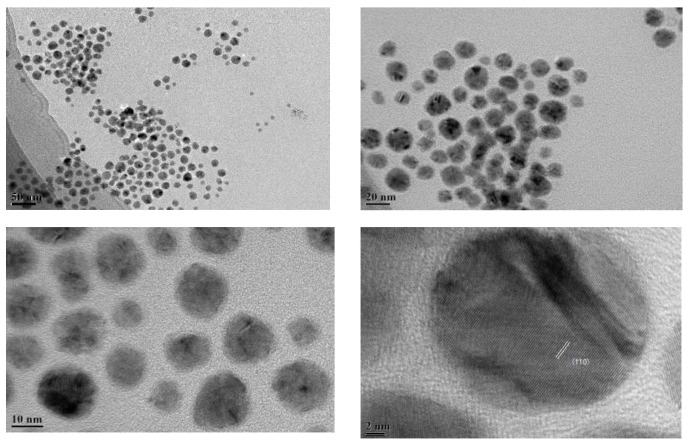
TEM of nanoparticle (Ni).

**Figure 3 nanomaterials-14-01877-f003:**
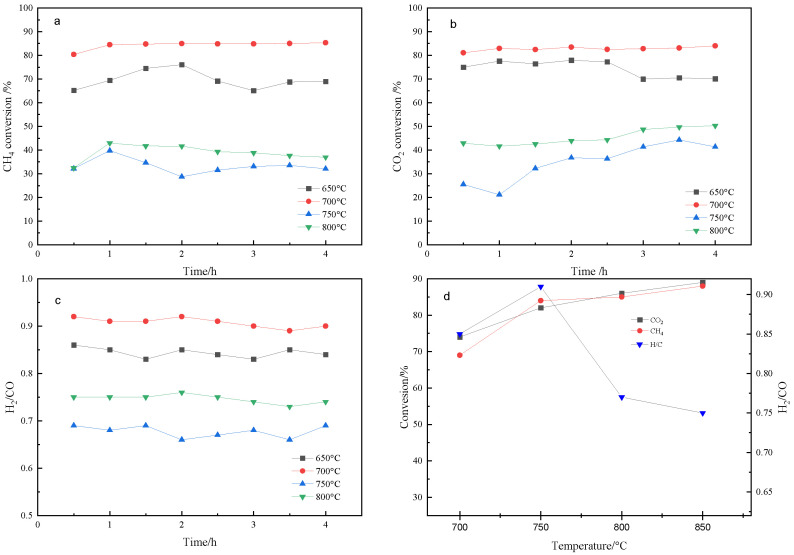
Effect of pretreatment temperature and temperature on catalyst performance (Ni@SiO_2_) ((**a**): CH_4_ of conversion, (**b**): CO_2_ of conversion, (**c**): ration of H_2_ and CO, (**d**): effect of reaction temperature).

**Figure 4 nanomaterials-14-01877-f004:**
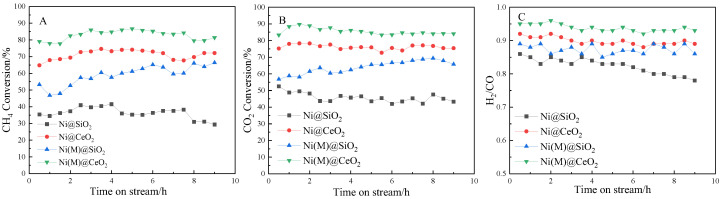
Effects of core and shell of catalysts on carbon deposit.

**Figure 5 nanomaterials-14-01877-f005:**
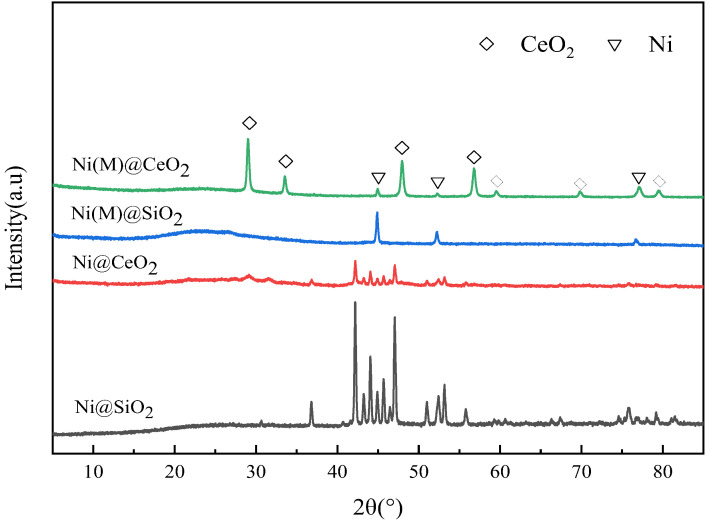
X-ray diffractograms of four reduced catalysts.

**Figure 6 nanomaterials-14-01877-f006:**
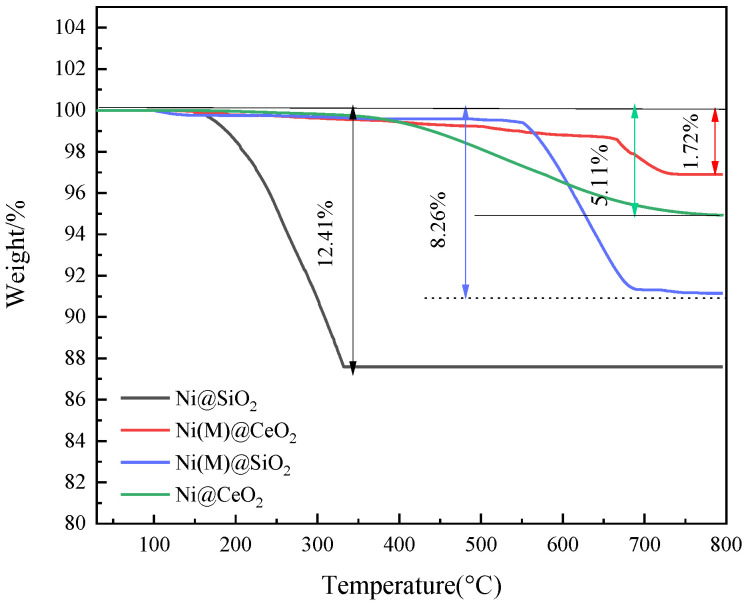
TGA of spent catalysts (Ni@SiO_2_, Ni(M)@SiO_2_, Ni@CeO_2_, and Ni(M)@CeO_2_).

**Figure 7 nanomaterials-14-01877-f007:**
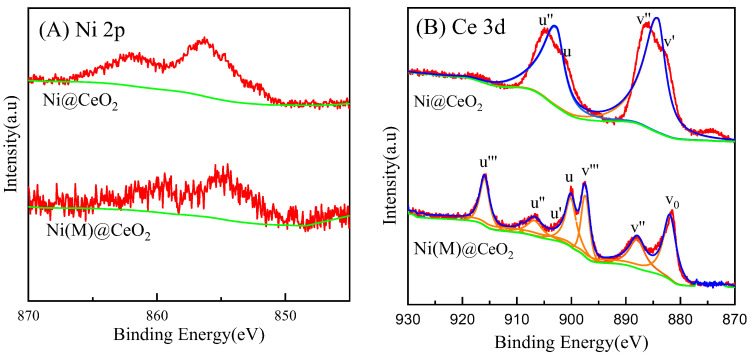
Ni 2p and Ce 3d XPS spectra of spent Ni@CeO_2_ and Ni(M)@CeO_2_ catalysts. ((**A**): Ni 2p, (**B**): Ce 3d).

## Data Availability

All relevant data are contained within this manuscript and its Appendix A.

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
