# Peer review of "Syngas from Reforming Methane and Carbon Dioxide on Ni@M(SiO_2_ and CeO_2_)"

_nanomaterials, 2024, doi:10.3390/nano14231877_

Round 1
Reviewer 1 Report
Comments and Suggestions for Authors
Synthesis gas is a valuable raw material and is used in the chemical industry for the production of methanol, which is a precursor to acetic acid and many acetates; liquid fuels and lubricants; ammonia via the Haber process, which converts atmospheric nitrogen (N2) into ammonia used as a fertilizer; oxo alcohols via an intermediate aldehyde and other valuable reagents. In this paper, the authors investigate the process of producing synthesis gas from methane and carbon dioxide, which are greenhouse gases. This process, in addition to producing synthesis gas, allows the utilization of greenhouse gases, so this study is important and relevant.
The authors have done a lot of work, but there are inaccuracies:
1.Page 1, line 32, missing the space between the word Figure and the number of the figure
2.Page 2, line 35. The authors refer to the equation of dry reforming of methane, but the equation itself is not given
3.Page 5, line 145-146. The space between the word Picture and number 3 is missing
4.Page 5, line 147-148. Sentences start with a small letter.
5.Page 5, line 153. The authors refer to reaction 2, but reaction 2 itself is not given
6.Page 5, line 156. The space between the word Picture and number 4 is missing
7.Page 5, line 159. Extra comma after Ni@CeO2(M)
8.Page 5, line 163. The authors refer to equation (2). The equation itself has not been found
9.Page 6, line 167-168. Provide links to a description of the deactivation of the catalyst due to sintering and coking
10.Page 7, line 198. The space between the word Picture and number 6 is missing
11.Page 7, line 199. Page 7, line 199. Incorrect degree designation
12. Page 7, line 207-208. The space between the word Picture and number 6 and is missing
13. in the Conclusion section, indicate what in your opinion are the reasons for the increased catalytic activity of cerium-containing catalysts and what yields and selectivity have been achieved with them
The manuscript was prepared inattentively. It is necessary to correct these remarks.
Author Response
Synthesis gas is a valuable raw material and is used in the chemical industry for the production of methanol, which is a precursor to acetic acid and many acetates; liquid fuels and lubricants; ammonia via the Haber process, which converts atmospheric nitrogen (N2) into ammonia used as a fertilizer; oxo alcohols via an intermediate aldehyde and other valuable reagents. In this paper, the authors investigate the process of producing synthesis gas from methane and carbon dioxide, which are greenhouse gases. This process, in addition to producing synthesis gas, allows the utilization of greenhouse gases, so this study is important and relevant.
The authors have done a lot of work, but there are inaccuracies:
- Page 1, line 32, missing the space between the word Figure and the number of the figure
The space was added into between Figure and 1(Fig.1 to Figure 1)
- Page 2, line 35. The authors refer to the equation of dry reforming of methane, but the equation itself is not given
the equation had been given.
- Page 5, line 145-146. The space between the word Picture and number 3 is missing
The space was added into between Fig. and 3(Fig.3 to Figure 3)
4.Page 5, line 147-148. Sentences start with a small letter.
“then” was modified into “Then”.
- Page 5, line 153. The authors refer to reaction 2, but reaction 2 itself is not given
the equation had been given.
- Page 5, line 156. The space between the word Picture and number 4 is missing
The space was added into between Figure and 4 (Figure4 to Figure 4).
- Page 5, line 159. Extra comma after Ni@CeO2(M)
Extra comma has been deleted.
- Page 5, line 163. The authors refer to equation (2). The equation itself has not been found
Equation has been added.
- Page 6, line 167-168. Provide links to a description of the deactivation of the catalyst due to sintering and coking
Line 167-172 is redundant and deleted.
- Page 7, line 198. The space between the word Picture and number 6 is missing
The space was added into between Figure and 6 (Figure6 to Figure 6).
- Page 7, line 199. Page 7, line 199. Incorrect degree designation
◦C was modified into ℃
- Page 7, line 207-208. The space between the word Picture and number 6 and is missing
The space was added into between Figure and 7 (Figure7 to Figure 7).
- in the Conclusion section, indicate what in your opinion are the reasons for the increased catalytic activity of cerium-containing catalysts and what yields and selectivity have been achieved with them
The conversion of CO2 and CH4 was added.
The manuscript was prepared inattentively. It is necessary to correct these remarks.
Reviewer 2 Report
Comments and Suggestions for Authors
The authors investigated the dry reforming of the methane using Ni@M (SiO2 and CeO2. The manuscript can be accepted after clarifying the following comments.
1. In page 5, line 161 Ni@SO2 should be Ni@SiO2.
2. The catalyst preparation procedure was not mentioned properly in the manuscript.
3. The representation of catalyst system was different in text and figures. Please follow the uniformity for better understanding of the readers.
4. The oxygen vacancies of CeO2 must be confirmed by characterization techniques. I would suggest the authors to include the results and their correlation with catalytic activity in the revised version.
5. The authors highlighted the basicity of the catalyst systems but the characterization results were not in the manuscript.
6. Figure 5, the XRD pattern of Ni@SiO2 is quite different from the reported literature. Please verify and correct it.
Comments on the Quality of English LanguageEnglish can be improved.
Author Response
Reviewer 2:
The authors investigated the dry reforming of the methane using Ni@M (SiO2 and CeO2. The manuscript can be accepted after clarifying the following comments.
- In page 5, line 161 Ni@SO2should be Ni@SiO2.
Ni@SO2 was modified into Ni@SiO2
- The catalyst preparation procedure was not mentioned properly in the manuscript.
The catalyst preparation procedure was stated in detail in 3.1 Catalyst synthesis
- The representation of catalyst system was different in text and figures. Please follow the uniformity for better understanding of the readers.
The catalyst was tagged uniformly
- The oxygen vacancies of CeO2must be confirmed by characterization techniques. I would suggest the authors to include the results and their correlation with catalytic activity in the revised version.
The oxygen vacancies of CeO2 have been confirmed by XPS techniques, and their correlation with catalytic activity have been stated in our manuscript.
- The authors highlighted the basicity of the catalyst systems but the characterization results were not in the manuscript.
The basicity of the catalyst was cited according to literature.
- Figure 5, the XRD pattern of Ni@SiO2is quite different from the reported literature. Please verify and correct it.
The different from the reported literature was caused by the characterization instrument.
Reviewer 3 Report
Comments and Suggestions for Authors
Dear Authors! The manuscript should be major revised before publish.
First of all, there are too much borrowing: the iThenticate report shows about 29% of borrowing for this study.
Other comments see belows:
1) Here are some mistakes such as "oleyacid", "cerous nitrate" et al. that make some difficulties to read and analyse the text. Some abbreviation (such as TOP) does not spelled out.
2) Physicochemical properties of the samples are limited to XRD, TEM, XPS and TGA analysis. Its does not enough to describe behavior and activity of the heterogeneous catalysts in the target reaction. Why You ignore some key properties, such as specific area and sites properties in the manuscript? Please apply some extra techniques to characterise properties (for example BET, TPR, TPO analysis).
3) Why GHSV influence did not study for the catalyst? Did You assupted that it parameter is not important or Your setup has some limitation to change GHSV?
Comments on the Quality of English LanguageText need to be checked and corrected.
Author Response
Dear Authors! The manuscript should be major revised before publish.
First of all, there are too much borrowing: the authenticate report shows about 29% of borrowing for this study.
Other comments see belows:
- Here are some mistakes such as "oleyacid", "cerous nitrate" et al. that make some difficulties to read and analyse the text. Some abbreviation (such as TOP) does not spelled out.
These mistakes in our manuscript were revised.
- Physicochemical properties of the samples are limited to XRD, TEM, XPS and TGA analysis. Its does not enough to describe behavior and activity of the heterogeneous catalysts in the target reaction. Why You ignore some key properties, such as specific area and sites properties in the manuscript? Please apply some extra techniques to characterise properties (for example BET, TPR, TPO analysis).
We think that it is enough to describe behavior in our manuscript.
3) Why GHSV influence did not study for the catalyst? Did You assupted that it parameter is not important or Your setup has some limitation to change GHSV?
GHSV will be studied in kinetics, here GHSV is constant.
Round 2
Reviewer 2 Report
Comments and Suggestions for Authors
Satisfied with the answers given by the authors. Now the manuscript can be accepted in its present form.